# Antiosteoporosis Effects, Pharmacokinetics, and Drug Delivery Systems of Icaritin: Advances and Prospects

**DOI:** 10.3390/ph15040397

**Published:** 2022-03-24

**Authors:** Lifang Gao, Shuang-Qing Zhang

**Affiliations:** 1School of Public Health, Capital Medical University, 10 Youanmenwai Xitiao, Beijing 100069, China; lifanggao@ccmu.edu.cn; 2National Institute for Nutrition and Health, Chinese Center for Disease Control and Prevention, 27 Nanwei Road, Beijing 100050, China

**Keywords:** icaritin, antiosteoporosis, pharmacokineticcs, drug delivery systems

## Abstract

Osteoporosis is a systemic skeletal disorder affecting over 200 million people worldwide and contributes dramatically to global healthcare costs. Available anti-osteoporotic drug treatments including hormone replacement therapy, anabolic agents, and bisphosphonates often cause adverse events which limit their long-term use. Therefore, the application of natural products has been proposed as an alternative therapy strategy. Icaritin (ICT) is not only an enzyme-hydrolyzed product of icariin but also an intestinal metabolite of eight major flavonoids of the traditional Chinese medicinal plant *Epimedium* with extensive pharmacological activities, such as strengthening the kidney and reinforcing the bone. ICT displays several therapeutic effects, including osteoporosis prevention, neuroprotection, antitumor, cardiovascular protection, anti-inflammation, and immune-protective effect. ICT inhibits bone resorption activity of osteoclasts and stimulates osteogenic differentiation and maturation of bone marrow stromal progenitor cells and osteoblasts. As for the mechanisms of effect, ICT regulates relative activities of two transcription factors Runx2 and PPARγ, determines the differentiation of MSCs into osteoblasts, increases mRNA expression of OPG, and inhibits mRNA expression of RANKL. Poor water solubility, high lipophilicity, and unfavorable pharmacokinetic properties of ICT restrict its anti-osteoporotic effects, and novel drug delivery systems are explored to overcome intrinsic limitations of ICT. The paper focuses on osteogenic effects and mechanisms, pharmacokinetics and delivery systems of ICT, and highlights bone-targeting strategies to concentrate ICT on the ideal specific site of bone. ICT is a promising potential novel therapeutic agent for osteoporosis.

## 1. Introduction

Osteoporosis, a systemic skeletal disorder, is caused by excessive bone resorption over bone formation, characterized by decreased bone mineral density, microarchitectural deterioration, increased bone fragility and susceptibility to fracture [1,2]. Worldwide, over 200 million people are estimated to suffer from osteoporosis, in which women over the age of 50 or postmenopausal women are four times more likely to develop the disease than men [3]. Annually, osteoporosis causes more than 8.9 million fractures throughout the world, resulting in annual costs of more than 10 billion dollars [4].

*Epimedium* (Berberidaceae) is an important traditional Chinese medicinal plant and has long been used alone or in combination with other herbs for the treatment of various diseases, including osteoporosis, tendon health, cardiovascular diseases, sexual dysfunction, and menstrual irregularity [5]. There are more than 260 compounds identified from *Epimedium* including 141 flavonoids, 31 lignins, 12 ionones, 9 phenol glycosides, 6 phenylethanoid glycosides, 5 sesquiterpenes, and other types of moieties, of which flavonoids are the major components and important chemotaxonomic markers [6]. Icariin is the most abundant constituent and accounts for more than 5.0% of the dried weight of an alcoholic decoction of *Epimedium* [7]. Icaritin (ICT, Figure 1A) is not only a bioactive compound enzyme-hydrolyzed from icariin but also an intestinal metabolite of eight major flavonoids of *Epimedium* [8,9]. It exerts broad therapeutic capabilities such as osteoprotective effect [10], neuroprotective effect [11], cardiovascular protective effect [12], anti-cancer effect [13], anti-inflammation effect [14], and immune-protective effect [15]. Unfortunately, anhydroicaritin (Figure 1B) and wushanicaritin (Figure 1C) were regarded as ICT by mistake in some reports [6,16] possibly as they had similar chemical structures. On 10 January 2022, ICT was approved for the treatment of advanced hepatocellular carcinoma by China National Medical Products Administration. ICT is currently undergoing phase 1 clinical trial for the treatment of osteoporosis (ClinicalTrials.gov Identifier: NCT02931305). ICT targets osteogenesis pathways in mesenchymal stem cell, osteoblast, and osteoclast cell lineages, and displays beneficial effects on bone health in osteoporosis animal models [17]. Particularly, the prominent osteogenic effects of ICT made it a promising anti-osteoporotic drug candidate since ICT as a natural phytoestrogen may negate the high risks of hormone replacement therapy in clinic [8]. However, unfavorable intrinsic physicochemical and pharmacokinetic properties of ICT restrict its anti-osteoporotic effects, therefore, various novel drug delivery systems have been developed to dissolve the problems. Over the past decades, few literature reviews and book chapters have involved the topic. Therefore, osteogenic effects and mechanisms, pharmacokinetic properties and delivery systems of ICT are reviewed and discussed.

## 2. Effects of Icaritin on Osteoporosis

Healthy bone, a dynamic living tissue, is mainly composed of two distinct types of cells: (1) osteoblasts, derived from bone marrow mesenchymal stromal cells (MSCs), which are bone-forming cells, and (2) osteoclasts, derived from bone marrow hematopoietic progenitors, which are multinucleated bone-resorbing cells. If the dynamic balance between bone formation and resorption is destroyed, bone metabolism disorders occur such as osteoporosis and osteopetrosis [18]. There are many factors that promote the occurrence and progression of osteoporosis, but the fundamental mechanism is an imbalance between osteoblasts and osteoclasts, including: (1) decreased differentiation and activity of osteoblasts result in reduced bone deposition, and (2) increased osteoclasts differentiation and activity lead to excessive bone resorption [19]. Therefore, osteoporosis therapy focuses on rebuilding the balance between bone formation and resorption.

### 2.1. Mesenchymal Stromal Cells

In 1951, Lorenz and his coworkers first verified that bone marrow acted as a pool of hematopoietic stem cells (HSCs) to maintain blood cells homeostasis for lifespan in mice and guinea pigs experiments [20]. Further research found that adult bone marrow-derived multipotent stem cells contained not only HSCs but also non-hematopoietic cells, that is MSCs. Then, MSCs have also been isolated from several other parts of the body including the adipose tissue, umbilical cord blood, skin, and amniotic fluid [21,22]. A series of studies, both in vivo and in vitro, have shown that MSCs have distinct differentiation potential under special microenvironment [23,24]. Since 1995, MSCs have been used in clinically studied experimental cell therapy for a wide range of diseases, such as systemic sclerosis [25], neurodegenerative diseases [26], liver regeneration [27,28], osteoarthritis [29], osteonecrosis [30], angiogenesis [31], ischemic brain [32], etc.

Fundamentally, MSCs are precursors for both osteoblasts and adipocytes, between which an inverse relationship exists (Figure 2). Bone loss is commonly accompanied by increasing bone marrow adiposity, that means MSCs play an important role in the development of osteoporosis. Therefore, promoting MSCs to osteogenesis is beneficial to bone regeneration. Several research have shown some small molecules such as ascorbic acid, β-glycerophosphate, and dexamethasone can stimulate osteoblastogenesis of MSCs by increasing the activity of alkaline phosphatase (ALP), calcium deposition, and suppressing adipogenesis of MSCs [33,34,35]. In 2013, Sheng et al. first reported that the novel semisynthetic molecule ICT could promote osteogenic differentiation and suppress adipogenesis of MSCs [10].

Runx2 is a key transcription factor for skeletal mineralization due to its regulation of extracellular matrix genes such as ALP, osteopontin, and type I collagen [36,37]. PPARγ plays an essential role in regulating development of the adipose lineage [38], and the relative activities of two transcription factors Runx2 and PPARγ determine whether MSCs differentiate into osteoblasts or adipocytes [39] (Figure 2). Sheng et al. [10] further identified that ICT promoted osteogenic differentiation and maturation of MSCs through initiating activation of Runx2. Meanwhile, their investigation [10] showed that ICT inhibited adipogenesis of MSCs through suppressing PPARγ (Figure 2).

Runx2 is activated by estrogen receptor and mitogen-activated protein kinase (ERK/MAPK)-dependent phosphorylation [39,40]. ERK/MAPK signaling pathways are essential for cellular biochemical and physiological processes including cell proliferation, migration, and differentiation. Luo et al. demonstrated that ICT could activate ERK/MAPK signaling pathway and facilitate the orientation of osteogenic differentiation of bone marrow MSCs in vitro [41] (Figure 2).

Stromal cell-derived factor-1 (SDF-1)/cysteine I-X-C motif chemokine receptor 4 (CXCR4) axis is required for mobilization and recruitment of MSCs, as well as proliferation and survival of MSCs [42,43] (Figure 2). Using an in vitro cell culture model, Lim et al. demonstrated that ICT enhanced MSCs proliferation, chemotaxis to SDF-1, and osteogenic differentiation, through the activation of signal transduction activator transcription factor (STAT-3) [44].

### 2.2. Osteoblasts

Osteoblasts, originating from MSCs, mediate bone-formation of new bone tissue, which is coupled by osteoclast-mediated bone resorption of old bone tissue. Therefore, stimulation of osteoblasts proliferation and activation is a target for new bone-forming to prevent osteoporosis. Osteoprogenitors, osteoblasts, and osteocytes belong to osteoblast lineage cells. The mature osteoblasts, located on the bone surfaces, are responsible for the synthesis and mineralization of the organic matrix rich in type I collagen and osteocalcin in the initial bone formation phase [45,46]. Early osteoprogenitors express Runx2 and osterix which are two critical transcription factors for osteoblasts differentiation and function [45,47,48,49,50] (Figure 2). CXCR4 is critical for maintaining osteoblast anabolic function. ICT enhanced the differentiation of MC3T3-E1 preosteoblastic cells and subsequently resulted in mineralization, collagen synthesis, and bone formation, because ICT promoted mRNA and protein expression of bone-forming biomarkers, such as ALP, type I collagen, osteocalcin, OPN, and Runx2 [51]. Further study revealed the mechanisms might be associated with ERK/MAP signaling pathway activated by ICT [51]. Wei et al. found that ICT could promote maturation and mineralization of MC3T3-E1 cells through SDF-1/CXCR4 signaling pathway in a series of in vitro experiments [52]. Lim et al. examined whether ICT could increase human osteoblast anabolic function. Both cellular and animal experiments showed that ICT increased osteoblasts proliferation and function. The underlying mechanism was that ICT suppressed the phosphorylation of STAT-3 to upregulate CXCR4 expression [53]. Peng et al. performed a study to examine the relationship between ICT treatment initiation time and bone turnover markers in adult ovariectomized rats, and found that early ICT treatment (1 month post-operation), not late ICT treatment (3 months post-operation), exerted beneficial effects on osteoporotic bone in ovariectomized rats [54]. Meanwhile, they performed a series of experiments to evaluate the population of osteoblasts with colony formation assays, assess the expression levels of osteoblasts-related genes by real-time polymerase chain reaction. Late ICT treatment failed to increase bone-forming related parameters [54].

Osteoblasts play an important role in maintaining bone homeostasis. Mature osteoblasts produce type I collagen and osteocalcin, and regulate activity of osteoclasts through receptor activator of nuclear factor-κB ligand (RANKL) and osteoprotegerin (OPG) [45] (Figure 2). Currently, OPG/RANKL/RANK signaling pathway is a crucial signal pathway for bone remodeling for the interaction between osteoblasts and osteoclasts, as well as a major pathway for affecting bone metabolism and for preventing and treating osteoporosis [55,56,57]. The trimolecular complex belongs to the superfamily of tumor necrosis factor (TNF) [58]. The receptor activator of NF-κB (RANK) is located on the osteoclasts surface, and its major ligands are OPG and RANKL. OPG, a secreted glycoprotein, has been identified to regulate bone resorption [57,59]. RANKL is mainly derived from osteocytes, osteoblasts, and MSCs during bone remodeling [58], and RANK is the only known receptor for RANKL. OPG and RANKL compete for RANK, which determines the balance between bone remodeling and resorption [60]. Huang et al. demonstrated that *Epimedium*-derived prenylflavonoids including icariin, icariside II, and ICT could promote proliferation, alkaline phosphatase activity, osteocalcin secretion of osteoblasts, and matrix mineralization [61]. Furthermore, the experiment demonstrated that ICT was more potent than other extracts, because ICT increased mRNA expression of OPG, and inhibited mRNA expression of RANKL [61].

### 2.3. Osteoclasts

Osteoclasts, arose from bone marrow hematopoietic monocyte/macrophage progenitors, mediate bone resorption of old bone tissue. The regulation of osteoclasts differentiation and activation involve signaling induced by RANKL and its receptor RANK [62,63,64,65,66,67] (Figure 2). Once RANK is activated by RANKL, osteoclasts differentiate and promote bone resorption. However, excessive activation of RANK/RANKL signaling pathway leads to osteoporosis. Therefore, inhibition of the RANKL-induced osteoclasts formation is an effective therapy for osteoporosis. TNF receptor-associated factor6 (TRAF6), a critical adaptor protein, is necessary for bone resorption [68], which is supported by several studies that TRAF6-deficient mice generated osteopetrosis [69,70,71]. Liu et al. evaluated the growth inhibitory effect of ICT on preosteoclastic RAW264.7 cells, and found ICT suppressed osteoclastic differentiation and activity in a dose-dependent manner [72]. Furthermore, Tan et al. revealed that ICT suppressed osteoclastogenesis in two osteoclast precursor models, RAW 264.7 mouse monocyte cell line and human PBMC, through inhibition of RANK/RANKL and MAPK/AP-1 signaling pathways and promotion of proteasomal degradation of TRAF6 [73]. A randomized, double-blind, placebo-controlled trial showed that ICT increased the bone anabolism marker such as bone specific alkaline phosphatase and suppressed TRAF6 protein in peripheral blood osteoclast-precursor monocytes in post-menopausal women [74].

### 2.4. Inflammation and Osteoporosis

Recently, more attention has been paid on the relationship between inflammation and osteoporosis. Evidence has shown that inflammatory cytokines play an important role in the osteoporosis processing of hormonal deficiency-induced rat model [75,76]. Accumulating data suggested that pro-inflammatory cytokines such as interleukins (IL), tumor necrosis factor-alpha (TNF-α), chemokines, interferons could induce osteoclastic bone resorption [77,78]. In 2013, Lai et al. first demonstrated the anti-inflammatory effect of ICT in lipopolysaccharide (LPS)-induced mouse peritoneal macrophages in vitro and peritonitis model in vivo [14]. In this report, the researchers found pretreatment of ICT significantly could inhibit the inflammatory cytokines production, including IL-6, IL-10, MCP-1, IFN, TNF, and IL-12p70 [14]. Then anti-inflammatory activity of icariin and its metabolites have been widely reported in different areas of disease [79,80,81].

### 2.5. Animal Bone Defect Model

Based on the findings that ICT might be a therapeutic small molecule agent for bone reconstruction, a series of animal bone defect model experiments systematically evaluated bone regeneration of bioactive scaffolds incorporating phytomolecule ICT which served as exogenous growth factor [82,83,84]. Not only in the non-loading bearing mechanical stresses rat calvarial defects model, but also in a standard rabbit ulnar segmental defect model, the bioactive scaffold incorporating ICT enhanced newly mineralized bone and new vessel growth. In conclusion, as an exogenous growth factor, ICT is beneficial to bone regeneration.

## 3. Pharmacokinetics of Icaritin

For the exploration of ICT pharmacokinetics, ICT in human and rat serum was determined by GC-MS and LC-MS/MS following the oral administration of *Epimedium* extract containing ICT, icariin, desmethylicaritin, icariside I, icariside II, epimedin A, epimedin B, and epimedin C [85,86]. Unfortunately, pharmacokinetic parameters failed to be shown because other components of *Epimedium* extract were biotransformed to ICT [87], resulting in big errors on calculating pharmacokinetic parameters. Although pure ICT was orally administered to rats, ICT failed to be quantified in plasma, bile, and urine due to low quantification of HPLC-UV [88]. For the intravenous administration of pure ICT at a single dose of 5 mg/kg, most of ICT at various sampling time-points could not be detected, therefore, calculated pharmacokinetic parameters were doubtful [88].

The in vitro metabolism investigation of ICT was performed in pooled human liver microsomes, pooled rat liver microsomes, pooled human intestine microsomes, and UDP-glucuronosyltransferase enzymes (UGT) [89,90,91,92]. ICT showed potent inhibitive effects on the activities of CYP1A2, CYP2C9, and CYP3A4 [91], as well as UGT1A1, UGT1A3, UGT1A7, UGT1A9, and UGT2B7 [89,90,92], suggesting the co-administration of ICT with the substrates of these enzymes should be avoided. Four ICT glucuronides, i.e., icaritin-3-glucuronide (G3), icaritin-5-glucuronide (G5), icaritin-7-glucuronide (G7), and icaritin-3,7-diglucuronide (G37), were identified and quantified in microsomal incubation systems, rat intestinal perfusion and portal vein infusion [89,92,93]. ICT underwent mono-glucuronidation by the intestine and subsequent was transported to the liver by organic anion transporting peptides, followed by biliary excretion mainly as diglucuronide [93]. Although monoglucuronide is occasionally metabolized to diglucuronide, diglucuronide was found to be the most abundant metabolite of ICT, both in vivo and in situ [93], which was not totally consistent with in vivo studies [9,94,95,96,97,98]. In rats following the oral administration of ICT, a total of 24, 25, and 30 metabolites were identified separately in different experiments, and besides glucuronidation, other metabolite types were demethylation, methylation, dehydrogenation, epoxidation, hydroxylation, dihydroxylation, reduction, sulfation, oxidation, and acetylation, in which dehydrogenation at isopentenyl group and glycosylation and glucuronidation at the aglycone were the major metabolism processes [9,97,98]. In particular, ICT was oxidized to a transient quinone methide intermediate in human microsomes, which may induce hepatotoxicity or may favor a favorable antioxidant activity, and then reacted with glutathione or N-acetyl-L-cysteine to nontoxic conjugates [99]. In our investigations [9,95], high polar conjugate G7 was the major metabolite accounting for 52.7% of all 24 metabolites, predominantly distributed in the kidney and slowly eliminated in the urine (a half-life of 4.5 h) representing 63.28% of absorbed ICT in rats.

Several pharmacokinetic studies were carried out to investigate the absorption, distribution, metabolism, and excretion of ICT in animals [5,8,9,93,94,95,98]. The major pharmacokinetic parameters are presented in Table 1. Due to the poor aqueous solubility (<0.2 μg/mL) and high lipophilicity (logP 6.7) of ICT, only 4.33% ICT was rapidly absorbed into the blood in rats after the oral administration, and subsequently the absorbed ICT was rapidly and predominantly metabolized into G7, of which plasma concentration and the area under the curve (AUC) were about eight-fold higher than those of ICT [9]. Approximately 65.7% of absorbed ICT was distributed in the liver for the oral route [9], and G7 was mainly distributed in the kidney and liver for both oral and intraperitoneal routes [94]. Interestingly, the excretion profile of ICT depends on the administration route. For the oral administration, most absorbed ICT was excreted as GICT via the urine and unabsorbed ICT was mainly excreted as the parent form in feces [9]. For the intravenous route, most ICT was excreted as conjugated metabolites via the bile [88]. ICT was slowly eliminated from plasma and various tissues (liver, spleen, kidney, heart, lung, muscle, and brain) with t_1/2,λz_ values of 6.64–20.92 h for the oral route [9,98] and 3.14–47.81 h for the intraperitoneal route [94,95].

## 4. Drug Delivery Systems

An ideal therapy for bone diseases would allow effective concentration of a low dose of therapeutic agents in the bone, thereby maximizing the local therapeutic index of drugs while minimizing the adverse effects at non-skeletal sites. Generally, bone consists of 50–70% inorganic mineral, 20–40% organic matrix, 5–10% water, and 1–5% lipids, in which hydroxyapatite constitutes the major component of inorganic mineral [1,100]. The complex microenvironment, structure, and architecture of the bone are the main obstacles for delivering therapeutic agents to bones via conventional administration modes. High mineral contents, deficiency of natural biological target receptors for most bone diseases, low-blood flow, and poor permeability are other obstacles [1]. The bone-formation surface covered with osteoblasts is mainly lowly crystallized hydroxyapatite and amorphous calcium phosphonate whereas the bone-resorption surfaces covered with osteoclasts is mainly highly crystallized hydroxyapatite [100,101]. The treatment of bone-related diseases with non-targeted drug delivery is a great challenge as it requires higher drug doses in order to reach effective drug concentrations in the bone tissue, which induces non-skeletal toxicity. Moreover, non-targeted drug delivery also induces insufficient therapeutic efficacy due to poor distribution of the molecule to avascular bone region [102]. Therefore, it is critical to develop safe and effective drug delivery systems to concentrate drug on the bone disorder site. Nanoparticle-based targeted drug delivery system exhibits prominent potential in osteoporosis treatment due to its excellent physicochemical properties, such as enhancing drug solubility and loading capacity, retarding degradation of drug molecules, and favoring manipulation of pharmacokinetic characteristics of drugs, thereby achieving high bioavailability and minimal toxicity. In terms of composition materials, there are two categories of nanoparticles, namely hard and soft particles. Soft nanoparticles consist of liposomes, dendrimers, micelles, and polymeric particles, and hard nanoparticles contain inorganic and metallic particles, such as silica and gold, quantum dots and carbon nanotubes [103]. Modifications of size, shape, surface charge, and porosity of nanoparticles effectively control their cellular uptake and biodistribution. In the light of delivery mode, targeted delivery is divided into two categories: passive and active targeting. In the bone, passive targeting takes advantage of bone-marrow capillaries’ fenestrations, and subsequently nanocarriers are captured by the macrophages and accumulate in bone tissues. Active targeting selectively accumulates in expected bone sites by virtue of ligand-receptor binding or specific interactions with osteoid [1]. For bone targeting, anionic ligands are usually employed to chelate superficial calcium ions on the resorption surface [100]. So far, many bone-targeting ligands including tetracyclines, bisphosphonates, oligopeptides, and aptamers have been developed [104,105,106,107,108] for specific bone sites, such as, tetracyclines [109], bisphosphonates [110], carboxylic acids [111], and peptide VTKHLNQISQSY [112] for all skeletal tissues due to their high affinity for hydroxyapatite, (AspSerSer)_6_ for bone-formation surface [101], (Asp)_6_ [113,114] and (Asp)_8_ [115,116] for bone-resorption surface, an integrin αVβ3-osteopontin protein-protein interaction inhibitor IPS-02001 for osteoclasts [117], Ser-Asp-Ser-Ser-Asp (SDSSD) [118] and its aptamer CH6 [119] for osteoblasts. To improve the selective delivery to particular cells or tissues, targeting ligands bind to the surface of active targeting systems. Two innovative approaches for the delivery of drugs to the bone are proposed: local implantable drug delivery system (local drug administration at a specified site) and targeted systemic drug delivery system (administration of drug into the circulatory system) [1]. Implantable bone graft substitutes, scaffolds, hydrogels, cement, and bone implant coatings are often employed for the delivery of therapeutic agents in local implantable drug delivery approaches. Passive or active bone-targeted delivery of ICT to the bone is a promising treatment for various bone disorders such as osteoporosis, osteoarthritis, osteonecrosis, bone fracture, osteosarcoma, and bone metastasis [1]. Several ICT delivery systems were developed including liposomes [115,120,121], micelles [122,123,124], nanoparticles [125,126], nanorods [127], nanocrystals [128], hydrogels [129], and bioactive scaffolds [82,83,84,130,131,132,133,134].

### 4.1. Liposomes

Liposomes are self-assembled small spherical colloidal vesicles consisting of amphiphilic phospholipids and cholesterol molecules forming enclosed bilayer structure that simulates the phospholipids of the human cell membrane, and the lipid vesicles could be encapsulated with both hydrophilic and hydrophobic drugs [100]. Among the nanocarriers, liposomes are the most clinically accepted drug delivery systems that are widely applied to deliver drugs, genes, vaccines, and imaging agents. Moreover, liposomes effectively transported multiple drugs without disadvantages of multidrug-loaded nanoparticles including unsatisfactory synergistic effects and controlled release for more than one drug which led to overdose and toxicity [103]. Especially, new generation liposomes have been developed, such as archaesomes, niosomes, novasomes, transfersomes, ethosomes, virosomes, cryptosomes, emulsomes, vesesomes, genosomes, and bilosomes.

Liposomes-based bone-targeted delivery of ICT seems to be an attractive approach for the effective treatment of bone disease since the system could overcome multiple limitations associated with the available drug formulations such as chronic therapy, low bioavailability, poor bone targeting, and off-target side effects [102]. With 1,2-dipalmitoyl-*sn*-glycero-3-phosphatidylcholine, soybean lysophosphatidylcholine and DSPE-polyethylene glycol 2000 (PEG2000) as lipid materials, ICT and coix seed oil dual loaded multicomponent thermosensitive liposomes was prepared by thin film hydration method [121]. In vitro and in vivo therapeutic potentials of the liposomes for hepatocellular carcinoma were improved via the anti-angiogenesis strategy and comprehensive tumor microenvironment remodeling, including HIF-1α-VEGF downregulation, depletion of cancer-associated fibroblasts, and inhibition of M2-type tumor-associated macrophage infiltration in desmoplastic tumor. Using the lipids of 1,2-dioleoyl-3-trimethylammonium-propane, dioleoylphosphatidylethanolamine, cholesterol, DSPE-PEG2000, and DSPE-PEG2000-MAL, Chen et al. [115] developed an (Asp)_8_-modified ICT liposome by thin film evaporation method, and found that the liposome efficiently prevented steroids-treated rats from steroid-associated osteonecrosis (SAON), with remarkably decreased osteocytes apoptosis and osteoclatsogenesis, as well as increased osteogenesis. Compared to the control liposome without (Asp)_8_, more ICT was selectively delivered to the bone-resorption surface and its therapeutic effects were enhanced via bone-resorption inhibition, adipogenesis suppression, and bone-formation enhancement. Similarly, an (Asp)_8_-modified ICT liposome prepared using cholesterol, soybean phosphatidylcholine, and DSPE-PEG2000-MAL was an effective bone-resorption surface targeting delivery system to load ICT for facilitating and promoting its therapeutic effects on the prevention of estrogen depletion-induced osteoporosis in ovariectomized mice [120]. A potential limitation of (Asp)_8_-modified ICT liposome was that total cholesterol in serum and thrombus area in the bone marrow were increased by disturbance of metabolism followed by fat accumulation, and osteoporotic phenotype was even promoted in animals [115,120].

### 4.2. Micelles

Polymeric micelles are self-assembled nanosized core-shell structures composed of amphiphilic polymers, which are capable to hold hydrophobic drugs in the hydrophobic inner core of micelles, hydrophilic bioactive molecules in the hydrophilic outer shell of polymeric micelles, and intermediate polar in between the core and the shell. Micelles are generally formed in aqueous solution when the polymer concentration increases above a certain specific concentration known as critical micelle concentration or critical aggregation concentration. Preparation methods of micelles include direct dissolution, solvent casting, dialysis, nanoprecipitation, interfacial polymerization, exfoliatione-adsorption, and template synthesis. Three types of polymers are often used in the preparation of micelles, diblock copolymers including polystyrene and polyethylene glycol (PEG), triblock copolymers including polyethylene oxide (PEO), and graft copolymers including stearic acid and chitosan. Polymeric micelles are often used in drug delivery due to their superior characteristics, such as target specificity, controlled-release property, tissue-penetration capability, biocompatibility, biodegradability, low toxicity, high stability, and high drug loading capacity.

Several investigations are available on the effective role of ICT-loaded micelles with enhancing solubility, stability, and bioavailability of ICT for the effective treatment of various diseases [122,123,124]. PEG-based amphiphilic block copolymer micelles are biocompatible and biodegradable, and possess significant potential for improving drug solubility and stability. Yang et al. [122] prepared ICT-loaded spherical polymeric micelles using amphiphilic copolymers of polycaprolactone (PCL)-PEG via film dispersion and evaluated its cytotoxicity and cellular uptake as well as intracellular distribution in CAL27 human oral squamous cell carcinoma cell lines. PCL-PEG micelles with the mean diameter of 121.2 nm exhibited stable and slow release of ICT, safe and effective drug delivery in OSCC cells compared to free ICT. Shan et al. [123] synthesized methyl PEG (mPEG)-polylactic acid (PLA) and prepared mPEG-PLA/ICT micelles by solid dispersion method. The micelles increased the solubility of ICT in water from 1.0 μg/mL to 2.0 mg/mL, and controlled sustained release of ICT in 108 h. Compared to free ICT, the rat plasma AUC and brain concentration of the mPEG-PLA/ICT micelles increased by 4.3 and 10 times, respectively. Rats pretreated with mPEG-PLA/ICT micelles can decrease neurological deficit score, diminish the infarct volume and brain edema. Tang et al. [124] developed mixed polymeric micelles with triblock copolymer poloxamer 407 and graft copolymer polyvinyl caprolactam-polyvinyl acetate-PEG (Soluplus^®^) using acid-base shift method to improve loading capacity and oral bioavailability of ICT. The micelles increased the bioavailability of ICT in beagle dogs by 14.9 times compared to ICT oil suspension.

### 4.3. Nanoparticles, Nanorods, and Nanocrystals

Nanostructures or nanocarriers, such as nanoparticles, nanorods, nanocrystals, nanofibers, nanospheres, nanoplates, nanowires, nanoflowers, nanoleaves, nanotubes, nanocages, nanofilms, nanosheets, nanochains, nanofoam, nanoholes, nanomesh, nanopillar, nanorings, nanoribbons, nanoshells, nanocubes, quantum wells, quantum dots etc., are rapidly developed due to the huge progress of nanomaterials [135].

According to the definition of the International Organization for Standardization, nanoparticles are discrete nano-objects with all three Cartesian dimensions of less than 100 nm, whereas drug nanoparticles have relatively large size (above 100 nm) for loading a enough amount of drug onto the particles [136]. Unique physical and chemical properties of nanoparticles are reliant on their size, shape, geometrical properties, and surface properties, which can all be modified through customized synthetic chemistry. Among the wide range of polymeric materials utilized to prepare nanoparticles, poly lactic-co-glycolic acid (PLGA) with favorable biocompatibility and degradability is widely used because PLGA nanoparticles reduce the therapeutic degradation and enhance the bioavailability of therapeutic agents with poor water solubility [1]. Generally, PLGA is modified with PEG to prolong its blood circulation by avoiding capture via the reticuloendothelial system [100]. Yu et al. [125] developed PLGA-PEG-aminoethyl anisamide nanoparticles to encapsulate both ICT and doxorubicin by solvent displacement method, characterized by the particle size of 138 nm and ICT encapsulation efficacy of 96.7%. Compared to free ICT intravenously injected to tumor-inoculated mice, the half-life time and the value of area under the curve of ICT nanoparticles significantly increased four-folds and three-folds, respectively. In hepatocellular carcinoma (HCC) mice, the nanoparticles remodeled the immunosuppressive tumor microenvironment and triggered a robust immune memory response, which significantly improved therapeutic effect to suppress early-stage HCC development. Tang et al. [126] designed amorphous ICT nanoparticles (64 nm) using Soluplus^®^ by reactive precipitation technique, and the oral bioavailability of ICT nanoparticles was 4.5-fold higher than that of clinical oil suspensions in beagle dogs.

Nanorods are rod-like nanosized objects composed of metals or semiconducting materials, and the easiest structures to prepare among all the nanostructures. Li et al. [137] formulated ICT nanorods by antisolvent precipitation method using D-α tocopherol acid polyethylene glycol succinate as a stabilizer. ICT nanorods with a diameter of 155.5 nm and a drug loading content of 43.3% exhibited excellent stability and a 7-day sustaining drug release manner. ICT nanorods showed significantly higher cytotoxicity than free ICT against MCF-7, 4T1, PLC/PRF/5, and HepG2 cancer cell lines. Nanorods improved in vivo anticancer efficacy of ICT in MCF-7 and PLC/PRF/5 tumor-bearing mice models.

Drug nanocrystals are nanoscale-sized pure drug crystals without any carrier material, and the dispersed nanocrystals are generally stabilized by low quantity of surfactants or polymeric stabilizers to prevent particle aggregation [138]. Nanosuspensions are submicrometer-scale colloidal dispersions of drug nanocrystals in external liquid media. Nanocrystals can be prepared by two different methods: bottom-up and top-down processes. Li et al. [128] fabricated ICT nanocrystals with a size of approximately 220 nm through the antisolvent precipitation method using hydroxypropyl methylcellulose as a stabilizer. However, the oral bioavailability was only enhanced two-fold compared with ICT suspensions.

### 4.4. Hydrogels

Hydrogels are three-dimensional cross-linked polymeric structures and can absorb and retain large amounts of water or biological fluids, which control the release of the drug loaded in hydrogels. Hydrogels are widely applied in biomedical engineering due to their tunable properties and their versatile preparation methods. Feng et al. [129] fabricated cell-infiltratable and injectable gelatin hydrogels by weak and highly dynamic host-guest complexations for the treatment of SAON. ICT and mesenchymal stem cells encapsulated in hydrogels were slowly released in two weeks, and ICT was delivered to boost differentiation of stem cells without the adverse effects associated with high drug dosage. The ICT-loaded hydrogels efficiently prevented the decreased bone mineral density, enhanced bone regeneration and facilitated the recruitment of endogenous cells to expedite the healing in the hip of SAON rat model.

### 4.5. Bioactive Scaffolds

Bioactive scaffolds incorporating bioactive compounds are bone tissue engineering scaffolds with three-dimensional structure, which provide necessary mechanical strength and repair bone defects. Synthetic and natural materials for bioactive scaffolds include polycaprolactone, PLA, PLGA, silk, collagen, hyaluronic acid, and chitosan [139]. Qin research group developed an ICT-releasing porous composite scaffold using PLGA/tricalcium phosphate by fine-spinning technology and evaluated the scaffold in vitro and in vivo [82,83,84,130,131,132,133,134]. Within 12 weeks, about 72% ICT was gradually released in the simulated body fluid, which promoted the proliferation and osteoblastic differentiation of rat bone marrow mesenchymal stem cells [131,134]. ICT scaffold significantly enhanced more ALP activity, upregulated mRNA expression of osteogenic genes and increased calcium deposition and mineralization in rabbit bone marrow stem cells [133]. After intramuscular implantation, ICT scaffold significantly promoted new bone formation within the bone defect in SAON rabbits and enhanced neovascularization in the rabbit muscle pouch model [83,131,132]. More newly formed bone and newly mineralized bone were observed in 12-mm ulnar bone defect rabbit model implanted with the scaffold [82]. Similarly, more area and volume fractions of newly formed bone were reported in calvarial defect rat model [84].

## 5. Conclusions and Future Perspectives

The review summarizes the recent advances and challenges faced in the osteogenic effects and mechanisms, pharmacokinetics and delivery systems of ICT for the treatment of osteoporosis. ICT inhibits bone resorption activity of osteoclasts and stimulates osteogenic differentiation and maturation of bone marrow stromal progenitor cells and osteoblasts. Furthermore, ICT regulates relative activities of two transcription factors Runx2 and PPARγ, determines the differentiation of MSCs into osteoblasts, increases mRNA expression of OPG, and inhibits mRNA expression of RANKL. Poor water solubility, high lipophilicity, and low bioavailability of ICT restrict its anti-osteoporotic effects, therefore, it is necessary to develop bone-targeted drug delivery systems to concentrate ICT on the bone disorder site. Liposomes, micelles, nanoparticles, nanorods, nanocrystals, hydrogels, and bioactive scaffolds are explored to selectively deliver ICT to bone by virtue of passive or active targeting strategy. ICT may exhibit the maximum anti-osteoporotic effects when ICT is specifically delivered to osteoblasts rather than osteoclasts.

## Figures and Tables

**Figure 1 pharmaceuticals-15-00397-f001:**
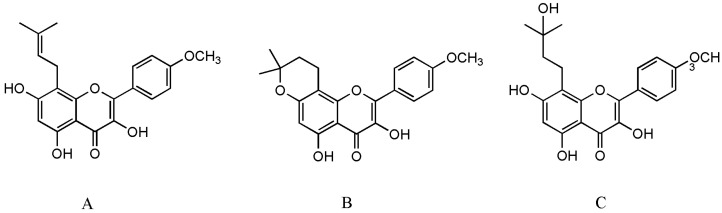
The structure of ICT (**A**), anhydroicaritin (**B**) and wushanicaritin (**C**).

**Figure 2 pharmaceuticals-15-00397-f002:**
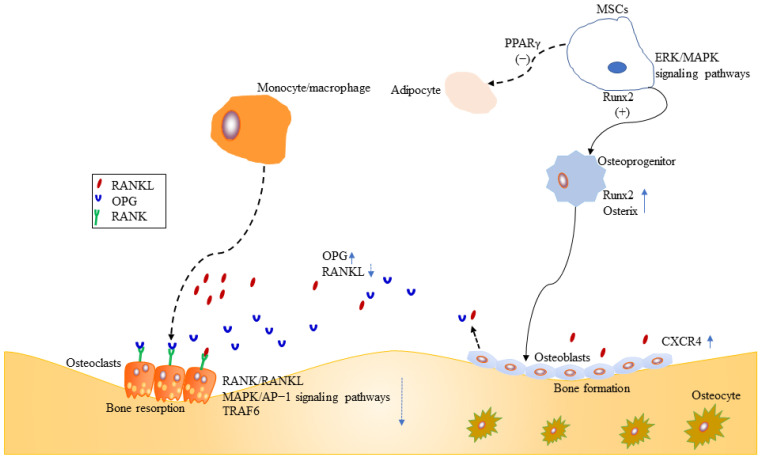
Effects of ICT on MSCs, osteoblasts and osteoclasts.

**Table 1 pharmaceuticals-15-00397-t001:** Pharmacokinetic properties of ICT in animals.

Animal	Dosage	Biological Sample	Quantification Method	Pharmacokinetic Properties	Reference
Male SD rats	5 mg/kg by single i.v.	Plasma	HPLC-UV	C_0_ (μg/mL) 7.20 ± 1.67 t_1/2,λz_ (h) 0.43 ± 0.11 AUC_0–2h_ (h·μg/mL) 1.42 ± 0.28 AUC_0–∞_ (h·μg/mL) 1.46 ± 0.30 Cl (mL/h/kg) 2.12 ± 0.42 V (L/kg) 3.54 ± 0.64	[88]
Female SD rats	20, 40, 60 mg/kg/day by i.p. for 7 days	Plasma	UPLC-MS/MS	20 40 60 mg/kg/day t_max_ (h) 1 0.5 1 AUC_0–8h_ (h·ng/mL) 1963 7450 15,885	[5]
FVB/NCrlVr mice	10 mg/kg by single i.p.	Spine	UHPLC-MS/MS	t_1/2,λz_ (h) 10.68 AUC_0–120h_ (h·ng/g) 642 Cl (g/h/kg) 15,486 V (g/kg) 238,683 MRT (h) 45.02	[8]
SD rats	10 mg/kg by single i.p.	Liver, spleen, kidney, heart, lung, muscle, adipose, and brain	UHPLC-MS/MS	t_1/2,λz_, t_max_, AUC_0–72h_, AUC_0–∞_, Cl, V, MRT_0–72h_, and MRT_0–∞_ of various tissues were obtained, see ref [94] for details	[94]
Male SD rats	2 and 40 mg/kg by single i.v. and i.g., respectively	Plasma	UHPLC-MS/MS	i.v. i.g. t_1/2,λz_ (h) 1.72 ± 0.37 7.37 ± 1.32 AUC_0–t_ (h·ng/mL) 699 ± 143 561 ± 97 AUC_0–∞_ (h·ng/mL) 700 ± 154 607 ± 89 Cl (L/h/kg) 2.86 ± 0.47 65.92 ± 11.38 V (L/kg) 7.10 ± 1.69 700.84 ± 107.95 MRT_0–t_ (h) 0.43 ± 0.12 6.02 ± 0.98 MRT_0–∞_ (h) 0.45 ± 0.14 8.18 ± 1.27	[9]
Male SD rats	40 mg/kg single i.g.	liver, spleen, kidney, heart, lung, muscle, and brain	UHPLC-MS/MS	t_1/2,λz_, t_max_, AUC_0–24h_, AUC_0–∞_, Cl, V, MRT_0–24h_, and MRT_0–∞_ of various tissues were obtained, see ref [9] for details	[9]
Male SD rats	100 mg/kg by single i.g.	Plasma	UPLC-MS/MS	t_max_ (h) 5.3 ± 1.1 C_max_ (ng/mL) 294.5 ± 22.7 t_1/2_ (h) 8.3 ± 1.0 AUC_0–48h_ (h·ng/mL) 3048.5 ± 289.0 AUC_0–∞_ (h·ng/mL) 3145.0 ± 302.3 MRT_0–48h_ (h) 9.6 ± 1.1 MRT_0–∞_ (h) 10.9 ± 1.3	[98]
Female SD rats	40 mg/kg by single i.p.	Plasma	UPLC-MS/MS	t_1/2,λz_ (h) 3.14 ± 0.34 AUC_0–24h_ (h·ng/g) 7937 ± 442 Cl (g/h/kg) 5.43 ± 0.85 V (g/kg) 22.79 ± 3.01 MRT_0–24h_ (h) 7.55 ± 0.97	[95]
Male Wistar rats	5 mg/kg by single i.g.	Plasma	UPLC-MS/MS	C_max_ (nmol/L) 7.12 ± 1.19 AUC_0–48h_ (h·nmol/L) 37.00 ± 12.98	[93]

## Data Availability

Data sharing not applicable.

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
