# Peer review of "Antiosteoporosis Effects, Pharmacokinetics, and Drug Delivery Systems of Icaritin: Advances and Prospects"

_pharmaceuticals, 2022, doi:10.3390/ph15040397_

Round 1

Reviewer 1 Report

The manuscript entitled “Antiosteoporosis effects, pharmacokinetics and drug delivery systems of icaritin: advances and prospects” is an interesting review describing how the icaritin could be used as potential therapeutic factor for osteoporosis.

The aims of the review are well described, and the general organization is well done. Anyway, it lacks a more specific description of the plant containing icaritin.

I suggest adding a new paragraph where you can describe the plant, its taxonomy, the parts of the plant used, what are the places where it grows, and so on. Furthermore, I suggest adding some photos about the plant or the utilized parts.

Author Response

Thank the reviewer for the comments. Initially, we wrote a new paragraph and put photos, as follows,

Epimedium, also known as barrenwort, bishop's hat, fairy wings, horny goat weed, or Yin Yang Huo in Chinese, is a genus of flowering plants in the Berberidaceae family. Currently, there are 63 acceptable species, of which the majority of the species are endemic to China, with smaller numbers elsewhere in Asia, and a few in the Mediterranean region. The genus was named by Carl Linnaeus in 1753, to describe the European species Epimedium alpinum. According to the Pharmacopoeia of the People's Republic of China 2020, Epidemidii Folium is the dried leaves of Epimedium Brevicornu Maxim., Epimedium Sagittatum Maxim., Epimedium Pubescens Maxim., or Epimedium Koreanum Nakai.

We though the paragraph over and over again, finally, we decide to remove it because (1) the manuscript focuses on icaritin not Epimedium, and (2) Epimedium was mentioned in paragraph 2, section Introduction.

Thanks again.

Reviewer 2 Report

The current manuscript is a well organized and valuable review on the osteogenic effects of icaritin, its pharmacokinetics as well as delivery systems, for the therapy of osteoporosis. Introduction provides the background and motivation of the study, while also presenting the potential of icaritin in the prevention and treatment of osteoporosis.

The following sections of the manuscript focus on the mechanisms by which icaritin acts on osteoporosis. Several publications are discussed that present the potential of ICT to induce osteogenic differentiation of MSCs, but also to regulate Runx2 and PPARγ transcription factors and act at the level of mRNA. The role of ICT in osteoblasts proliferation and function is also comprehensively discussed together with its role in maintaining a balance between bone remodeling and resorption. Moreover, the  antiinflammatory effect of ICT is presented and eventually the beneficial role of ICT to bone regeneration is concluded.

A very interesting section of the manuscript is the one describing the pharmacokinetics of ICT, by presenting the results of several studies that were centered around this topic.

Finally, this manuscript reviews the available drug delivery systems that are meant to overcome the main drawbacks of ICT use in therapy, like liposomes, micells, nanoparticles, hydrogels and scaffolds. All of them are critically discussed and the advantages of each of them is reviewed.

The manuscript ends with a pertinent conclusion and future challenges regarding this topic.

My opinion is that this review is very well written and also very much needed, since there are a lot of papers that tackled different aspects of ICT use but none to comprehensively gather all information in one place. 

Author Response

Thanks for your kind words!